# Association of missense variant *DCLRE1B* rs3761936 with breast and cervical cancer risk—A case-control study

**Sarah Jafrin**[1,2], **Md. Abdul Aziz**[1,2,3], **Md Abdul Barek**[1,2,3], **Md. Sharif Reza**[1],
**Nura Ershad Naznin**[1,2], **Mohammad Safiqul Islam**[1,2,3]*

**1** Department of Pharmacy, Faculty of Science, Noakhali Science and Technology University, Sonapur, Bangladesh, **2** Laboratory of Pharmacogenomics and Molecular Biology, Department of Pharmacy, Noakhali Science and Technology University, Noakhali, Bangladesh, **3** Bangladesh Pharmacogenomics Research Network (BdPGRN), Noakhali, Bangladesh

* research_safiq@nstu.edu.bd

## Abstract

### Background

Overexpression of rs3761936 of *DCLRE1B* gene has been observed in both breast cancer and cervical cancer patients. To justify the association of this polymorphism with these cancers, we performed this case-control study.

### Method

A total of 245 cancer patients and 108 healthy controls participated in the research. An efficient T-ARMS PCR method was used for genotyping.

### Results

The cancer patients showed higher mutant allele frequency compared to the controls. Mutant allele carrier breast cancer patients showed significantly increased risk in four genetic models, including additive model 1 (TC vs. TT: OR=2.31, 95% CI = 1.33–3.99, *p*-value = 0.0028), additive model 2 (CC vs. TT: OR=3.93, 95% CI = 1.36–11.38, *p*-value = 0.0116), dominant model (TC + CC vs. TT: OR=2.52, 95% CI = 1.50–4.25, *p*-value = 0.0005), and over-dominant model (TC vs. TT + CC: OR=1.93, 95% CI = 1.13–3.28, *p*-value = 0.0152). The allele frequency analysis showed that mutant allele C carriers among breast cancer patients had a significantly higher risk than the wild type T allele carriers (C vs. T: OR=2.15, 95% CI = 1.41–3.26, *p*-value = 0.0003). Likewise, the cervical cancer patients showed significant risk in three genetic models, including additive model 1 (TC vs. TT: OR=1.80, 95% CI = 1.01–3.20, *p*-value = 0.0444), additive model 2 (CC vs. TT: OR=3.17, 95% CI = 1.05–9.55, *p*-value = 0.0403), and dominant model (TC + CC vs. TT: OR=1.98, 95% CI = 1.15–3.41, *p*-value = 0.0138). The mutant allele C carriers had a significantly

**Data availability statement:** All relevant data are within the manuscript and its Supporting Information files.

**Funding:** The Research Cell, Noakhali Science and Technology University, Noakhali-3814, Bangladesh, funded this study partially (NSTU/RC/20/C-86), and no other public or private logistic funding was provided.

**Competing interests:** The authors have declared that no competing interests exist.

higher risk than the wild-type T allele carriers (C vs. T: OR=1.84, 95% CI = 1.19–2.85, *p*-value = 0.0065).

## Conclusion

*DCLRE1B* rs3761936 is strongly associated with breast cancer and cervical cancer risk in Bangladeshi women.

---

## Background

Cancer is an incessantly evolving critical health problem that results from the attenuation of multiple somatic and germline mutations [1]. Only in 2020, about 19.3 million new cases of cancer were registered, which caused 10 million deaths worldwide [2,3]. Genetic mutations are the inevitable facts in cancer development. Next-generation sequencing (NGS) has been playing a massive role in large-scale genomic discoveries and till now has revealed tremendous information about the fundamental genomic candidates that are directly or indirectly engaged in cancer development and progression [4]. Many recent genome-wide association studies have discovered a number of somatic and germline mutations in tumor suppressor genes, such as *TP53*, *RB1*, and *APC*. Some oncogenic mutations are identified as oncogenic activators from copy number variant analysis; for example- *HER2/ERBB2* and *MYC* have been identified and potentially targeted for molecular therapy [5–8]. Intense analysis of almost 50,000 cancer genomes has been conducted to evaluate multiple genomic alterations, variations, and their impact on cancer development and progression. Most of the common variants associated with cancer have already been vastly identified and analyzed [9,10]. Interest is now growing in the rare and newer variants of the driver genes for more specific findings on cancer genomics [6,11].

The most frequently diagnosed cancers among women worldwide are breast cancer (BC) and cervical cancer (CC). The mortality rate in women with cancer is highly variable. According to 2020 statistics, breast cancer was the leading cause of death among women in 110 countries, and the mortality rate was the highest for cervical cancer patients in 36 countries [12–16]. Breast cancer caused 6,846 deaths and 12,764 new cases in Bangladesh recently, while cervical cancer, the second leading cause of female death in Bangladesh, caused 5,214 fatalities and 8,068 new cases [17]. Till now, genome-wide association studies (GWASs) have discovered thousands of disease risk-associated single-nucleotide polymorphisms (raSNPs); among them, almost 150 variants are found to be susceptible to breast cancer and cervical cancer risk [18].

*DCLRE1B* gene is one of the vital members of the DNA cross-link repair family that remained evolutionarily conserved. The level of expression of *DCLRE1B* gene has been found to be significantly higher in the human brain (69.0). This gene encodes an essential protein referred to as '5' exonuclease Apollo' (60002 Da) structured by 532 amino acids that directly interact with *TERF2* in its quaternary form and partially interact with *MUS81*, *MRE11*, *FANCD2*, *HSPA2*, *HSPA8, HSPA14*, and *SPAG5*. The

exonuclease participates in the non-homologous end-joining (NHEJ)-mediated repair where *TERF2* directs 5'-3' exonuclease to form a telomeric loop (T loop) during replication and exposes the telomere end in a manner that activates the DNA repair pathways. This protein also facilitates double-strand break formation in response to DNA inter-strand cross-links. Diseases associated with *DCLRE1B* include Hoyeraal Hreidarsson Syndrome and Dyskeratosis Congenita [19–27].

The missense variant rs3761936 of *DCLRE1B* gene is located in the chr1:113907040 (GRCh38.p13) chromosomal anchor position. A missense mutation occurs when the change of a single base pair (bp) causes the substitution of a different amino acid in the resulting protein. This amino acid substitution may have no effect, or it may render the protein nonfunctional. A recent GTEx and ULCAN database showed significantly high expression of *DCLRE1B* mRNA in mammary breast tissue (Figs 1 and 2 (a), 2(b)). The UALCAN database also revealed the presence of significantly high expression of *DCLRE1B* mRNA in cervical cancer tissues compared to healthy tissues, which indicates a vital role of the *DCLRE1B* gene in the development of cervical cancer. As missense mutations tend to modify amino acid sequences, there remains a possibility of producing inactive mRNA or hyperactive mRNA. In both cases, normal expression of mRNA would not be observed if missense mutation is present. This low- or over-expression of mRNA may produce inactive or modified proteins that are ultimately associated with various diseases or conditions. In the case of rs3761936 of *DCLRE1B* gene, this missense mutation upregulates the mRNA expression in breast cancer and cervical cancer patients. There is a significant possibility that this specific polymorphism might play a potential role in breast and cervical cancer development [28–30].

Based on the prior studies, we chose the rare genetic variant of the *DCLRE1B* gene, a bi-allelic polymorphism rs3761936, that may influence breast cancer and cervical cancer development. To the best of our knowledge, no individual study has been performed on this polymorphism in any population except the GWAS study. Therefore, the aim of our present study is to evaluate the risk association of breast cancer and cervical cancer development with *DCLRE1B* rs3761936 polymorphism in Bangladeshi women.

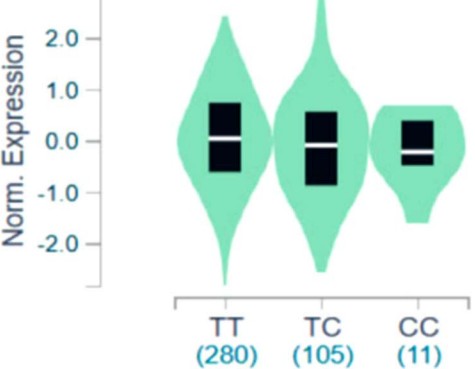

**Fig 1. GTEx database showing high expression of *DCLRE1B* mRNA in breast cancer vs. normal mammary tissue at the presence of rs3761936 polymorphism.**

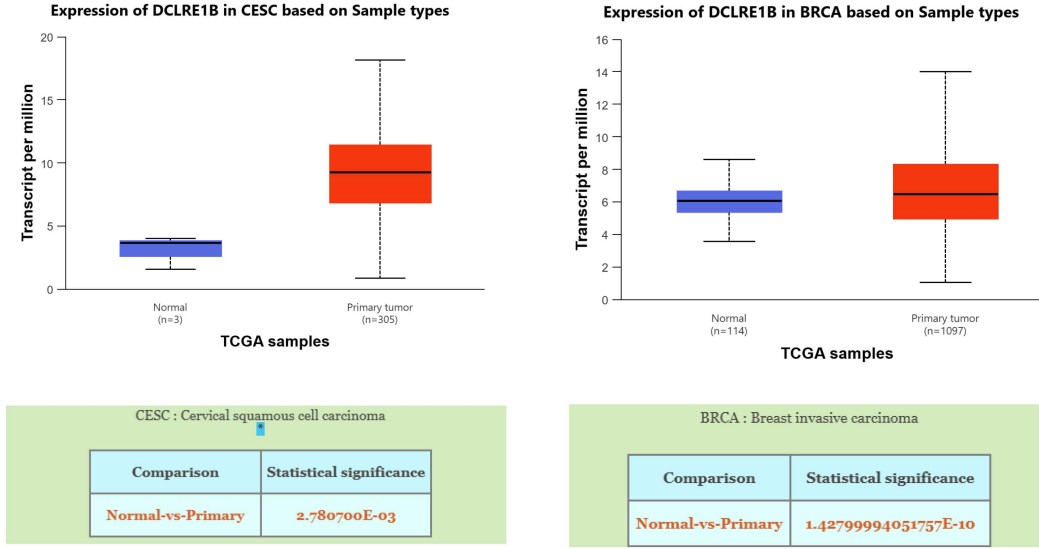

Fig 2. UALCAN database showing significantly high expression of *DCLRE1B* mRNA in both cervical cancer and breast invasive carcinoma tissue compared to healthy tissues.

## Method and materials

### Study setting

This case-control study was performed at the Laboratory of Molecular Biology and Pharmacogenomics of Noakhali Science and Technology University. The 1964 Helsinki Declaration and its later revisions, or equivalent ethical standards, were followed in all procedures conducted in research involving human subjects. The Ethical Review Board of the National Institute of Cancer Research and Hospital approved the study protocol (for breast cancer: NICRH/Ethics/2019/446 and for cervical cancer: NICRH/Ethics/2019/447). Patients' informed consent was obtained in advance. Three groups of participants were selected for the study. Two groups consisting of 135 women with breast cancer and 110 women with cervical cancer (aged 25–70 years) were recruited as cases from the National Institute of Cancer Research Hospital (NICRH), and 108 healthy volunteers (aged 21–74 years) from the volunteers who came to the hospital for their routine check-ups in NICRH were selected as controls. The blood samples were collected from July 1, 2019, to March 30, 2020. All the participants submitted their written informed consent prior to blood collection. We collected samples by following some inclusion and exclusion criteria. **Inclusion criteria:** Participants were all legally adults (above the age of 18) and genetically unrelated to others. The cases must have detailed diagnostics and medical history. Controls must be free from any severe disease and no previous family history of cancer. **Exclusion criteria:** Patients and controls without clear medical history were excluded from this study. We excluded the control population having serious illness or family history of cancers. Patients or controls under the age of 18 were also excluded from the experiment.

We collected demographic information from the participants like- age, BMI, marital status, lifestyle, clinical conditions and history of cancer. The genotypic information revealed from the blood samples- normal homozygote, heterozygote and mutant homozygote were used to determine the risk. The degree of heterogeneity between the demographic features of cases and controls was kept to a minimum. A standard questionnaire concerning disease history, as well as a genetic record of breast cancer in the family tree, were used to determine the patient's medical history. All the patients were also clear of other serious illnesses such as lung, renal, and liver disease. The information of the recruited control population revealed no evidence of any personal history of cancer or other long-term illnesses. The seventh edition of the American

Joint Committee on Cancer (AJCC) tumor-node-metastasis (TNM) staging system was used to classify breast cancer and cervical cancer.

## Sample collection, storage, and DNA extraction

For sampling and DNA extraction, 3 ml of blood was drawn from each patient and control participant, placed in sterile tubes with EDTA-Na$_2$, and stored at −80°C until DNA extraction. Following the extraction procedure described by Islam et al. [31], genomic DNA was isolated.

## Genotyping and screening

Utilizing all individuals' extracted DNA for genotyping, next we used T-ARMS PCR to amplify the target DNA fragments. Following the procedure described by previous studies from our lab [32,33], the SNP was genotyped. We designed four primers to detect the SNP in a DNA sample. Two forward outer and reverse outer primer were used to cleave and amplify the specific region of the gene that may or may not contain the polymorphism. Two allele specific primers were used that can identify the mutant allele and cleave the previously multiplied specific portion of the gene to produce DNA fragments of two different sizes. The mutant or wild type allele can be separated based on their size differences. This method does not require any digestion with restriction enzyme and less time consuming. Moreover, this method is comparatively more cost effective. Licensed software was used to create the primers. Our designed primer sequences for the gene are outlined on S1 Table. We used two thermal cyclers for polymerizing DNA collected from both patients and controls- SimpliAmp Thermal Cycler, Applied Biosystems, USA and SimpliAmp Veriti Thermal Cycler, Applied Biosystems, Taiwan. To find the desired polymorphism, agarose gel electrophoresis was used to examine amplified DNA fragments (Table 1).

## Method validation for selecting annealing temperature

The annealing temperature of a PCR is a crucial parameter that depends on the melting point of the primers. The temperature must be at least 5°C lower than the primers' melting point (Tm). We estimated Tm by utilizing Primer Blast software (S1 Table). The annealing temperature may deviate from the expected value because of buffer or salt content. To execute gradient PCR, we chose 6 different annealing temperatures between 59.5°C and 62°C and used our customized master mix formula for the procedure. We used two identical DNA samples at various temperatures to figure out the ideal temperature. After PCR, we used gel electrophoresis to observe the DNA bands of an individual allele. At 61°C, we detected 344 bp, 220 bp, and 180 bp size fragments, and their appearance was more pronounced than at the other

**Table 1. PCR condition to amplify the *DCLRE1B* rs3761936 and the respective length of PCR products.**

| SNP | PCR conditions | No. of cycles | Size of PCR products (bp) | Genotypes |
|---|---|---|---|---|
| *DCLRE1B* rs3761936 | 95°C for 5 minutes **Initial denaturation** | | | |
| | 95°C for 1 minute **Denaturation** | | NH: 220, 344 bp | TT |
| | 61°C for 45 seconds **Annealing** | 35 cycles | HE: 180, 220, 344 bp<br>MH:180, 344 bp | TC<br>CC |
| | 72°C for 1 minute **Extension** | | | |
| | 72°C for 10 minutes **Final extension** | | | |

\* NH- Normal Homozygote, HE- Heterozygote, MH- Mutant Homozygote.

temperatures. 61°C was therefore chosen as our preferred annealing temperature. We also reanalyzed 20% heterozygous genotypes to confirm the reliability of the genotyping.

## Statistical analysis

Two-tailed unpaired t-tests and the chi-square test ($\chi^2$) were used to examine the distributions of the demographic factors between the cases and the controls. We conducted t-tests on both cases and controls using their age and BMI as variables. The two-tailed $p$ values were 0.0001, extremely statistically significant for each variable. Standard error of differences was in the acceptable range- 1.351 and 1.081 respectably. The percentages of genotype and allelic frequencies were reported, and distribution of the alleles were evaluated by using Hardy-Weinberg equilibrium (HWE) test. The consistency of the allelic distribution is measured by chi-square value and $p$ value. We performed logistic regression analysis in three genetic models: additive, dominant, and recessive, to explore the association of mutant allele with the development of BC and CC in Bangladeshi population. To determine the strength of the link between this SNP and Bangladeshi breast cancer and cervical cancer patients, the odds ratio (OR) and their 95% confidence intervals (CI) were calculated by SPSS software package version 24.0 (SPSS, Inc., Chicago, IL). Results were deemed significant when the $p$-value was less than 0.05 and non-significant when the $p$-value was more than 0.05.

## Results

### Cases and controls characteristics

The distributions of demographic characteristics of 135 breast cancer patients, 110 cervical cancer patients, and 108 controls are summarized in Table 2. The age range of cancer patients was 25–70 and 21–74 for the control population. The average age for patients was 45.65 ± 11.66 years and for controls was 40.37 ± 11.77 years. The average body mass index (BMI) of patients was 29.39 ± 7.90 kg/m$^2$ and of controls was 22.30 ± 10.60 kg/m$^2$. This table also showed other characteristics, including- smoking habit, marital status, menstrual commencement, menopause, childbirth frequency, and breastfeeding status. Most of the patients (76.70%) first showed their symptoms above 40 years of age. 70.10% of patients had a history of taking contraceptive pills. Among the pill users, 52.13% of patients took pills for more than 5 years, and 47.87% of patients continued for less than 5 years, whereas 23 patients had a family history of different cancers.

Tables 3 and 4 show histopathological data of the BC and CC patients, respectively. According to BC patients' medical history, 62.96% of patients had invasive duct cell carcinoma, and 25.19% had infiltrating duct cell carcinoma, 3.70% patients had duct cell carcinoma, 2.22% had metastatic duct cell carcinoma, 1.48% had atypical ductal hyperplasia and intraductal carcinoma, and 2.96% had triple-negative breast cancer. TNM staging system reported the tumor size of the individuals. Nodal status data reported that most of the patients had N1 type tumor. We also included tumor grade, ER, PR, and HER2 status of breast cancer patients.

Of the 110 CC patients, 74.55% suffered from squamous cell carcinoma and 25.45% suffered from adenocarcinoma. According to tumor stage data, the majority of the patients had IIB type tumor. Cancer grading percentage was- grade 1 (26.36%), grade 2 (64.55%), and grade 3 (9.09%). Based on the tumor size analysis, 52.72%, 36.36%, 8.18%, and 2.72% of patients had T1, T2, T3, and T4 size, respectively. 84.61% of patients showed negative lymph node status as per our database. Nodal status was also included for the individuals and 52.63% had nodal status. According to distant metastasis data, patients were grouped into three groups- Mx (74.76%), M0 (22.33%), and MI (2.91%).

### Genotype and allele frequencies distribution

We performed the Hardy-Weinberg equilibrium (HWE) test for both patients and controls. Breast cancer patients had comparatively higher mutant homozygote and mutant allele frequency than in the control population. Based on our analysis of

**Table 2. Distribution of demographic variables of cancer patients and controls.**

| Variables | Cases n = 245 | Controls n = 108 |
|---|---|---|
| **Age (years)** | | |
| Minimum Age | 25 | 21 |
| Maximum Age | 70 | 74 |
| Average Age(mean±SD) | 45.65 ± 11.66 | 40.37 ± 11.77 |
| **BMI (kg/m2)** | | |
| Average(mean±SD) | 29.39 ± 7.90 | 22.30 ± 10.60 |
| **Marital Status** | | |
| Married | 238 (97.14%) | 93 (86.11%) |
| Unmarried | 7 (2.86%) | 15 (13.89%) |
| **Region of Patient** | | |
| Urban | 59.09% | 56.77% |
| Rural | 40.91% | 43.23% |
| **Menstruation Cycle Starting Age** | | |
| ≤13 | 68.18% | 28.82% |
| >13 | 31.82% | 71.18% |
| **Menopausal Age** | n = 150 | n = 50 |
| ≤45 | 80.12% | 24.22% |
| >45 | 19.88% | 75.78% |
| **First Child Conceived Age** | | |
| ≤18 | 75.45% | 60.49% |
| >18 | 24.55% | 39.51% |
| **Breastfeeding period** | | |
| <2 | 14.29% | 28.34% |
| ≥2 | 85.71% | 71.66% |
| **Commencement of Cancer Symptoms** | | |
| ≤40 | 23.30% | NA |
| >40 | 76.70% | NA |
| **History of Taking Contraceptive Pills** | n = 167 | n = 70 |
| Yes | 68.16% | 64.81% |
| No | 31.84% | 35.19% |
| **Smoking History** | None | None |
| **Family History of Cancer** | 20.82 | None |

\* NA- Not Applicable

*DCLRE1B* rs3761936 polymorphism in the BC patients, genotypic frequency distribution was consistent with the HWE test ($x^2$ = 0.02, *p-value* = 0.891). The frequency of the reference allele T was 80.09%, and mutant allele C was 19.91% in the control population ($x^2$ = 0.19, *p-value* = 0.664). Cervical cancer patients showed similar genotypic distribution to breast cancer patients. The frequency of the reference homozygote was higher in the control population than in the cervical cancer population. Genotypic frequency distribution was consistent with the HWE test ($x^2$ = 0.27, *p-value* = 0.601). Noticeable differences between the genotypic frequencies of the patients and controls indicate that *DCLRE1B* rs3761936 polymorphism might be associated with an increased risk of breast cancer (S2 Table).

**Table 3. Distribution of clinicopathological features of breast cancer patients.**

| Variables | Cases, n = 135 (%) |
|---|---|
| Histological Types of Breast Cancer | |
| Atypical ductal hyperplasia | 2 (1.48) |
| Duct cell carcinoma | 5 (3.70) |
| Infiltrating duct cell carcinoma | 34 (25.19) |
| Intraductal carcinoma | 2 (1.48) |
| Invasive duct cell carcinoma | 85 (62.96) |
| Metastatic duct cell carcinoma | 3 (2.22) |
| Triple negative breast cancer | 4 (2.96) |
| TNM Staging System | |
| Tumor size | |
| Tx | 0 |
| Tis | 0 |
| T0 | 27 (20.00) |
| T1 | 45 (33.33) |
| T2 | 34 (25.19) |
| T3 | 14 (10.37) |
| T4 | 15 (11.11) |
| Nodal status | |
| Nx | 25 (18.52) |
| N0 | 19 (14.07) |
| N1 | 59 (43.70) |
| N2 | 15 (11.11) |
| N3 | 17 (12.59) |
| Distant metastasis | |
| M0 | 102 (75.56) |
| M1 | 33 (24.44) |
| Grade of Breast Cancer | |
| I | 23 (17.03) |
| II | 89 (65.93) |
| III | 23 (17.03) |
| ER Status | |
| ER (+) | 56 (41.48) |
| ER (-) | 79 (58.52) |
| PR Status | |
| PR (+) | 57 (42.22) |
| PR (-) | 78 (57.78) |
| HER2 Status | |
| HER2 (+) | 60 (44.44) |
| HER2 (-) | 75 (55.56) |

## Association between breast cancer and rs3761936 polymorphism

We used six genetic models to quantify the risk of *DCLRE1B* rs3761936 polymorphism on breast cancer patients (Table 5). The risk was estimated by the odds ratios of each genetic model, and the control population was considered as a reference. For breast cancer patients, additive model 1 showed 2.31-fold significantly increased risk of cancer

**Table 4. Clinicopathological features of cervical cancer patients.**

| Variables | Cases, n = 110 (%) |
|---|---|
| **Type of Cancer** | |
| Squamous cell carcinoma | 82 (74.55) |
| Adenocarcinoma | 28 (25.45) |
| **Tumor Stage (N = 103)** | |
| I | 9 (8.74) |
| IIB | 52 (50.49) |
| IIIA | 3 (2.91) |
| IIIB | 36 (34.95) |
| IVA | 3 (2.91) |
| **Grade of Cancer** | |
| Grade 1 | 29 (26.36) |
| Grade 2 | 71 (64.55) |
| Grade 3 | 10 (9.09) |
| **Tumor Size** | |
| T1 | 58 (52.72) |
| T2 | 40 (36.36) |
| T3 | 9 (8.18) |
| T4 | 3 (2.72) |
| **Lymph Node Status (N = 104)** | |
| Negative (-) | 88 (84.61) |
| Positive (+) | 18 (17.30) |
| **Nodal Status** | |
| N1 | 10 (52.63) |
| N2 | 8 (42.11) |
| N3 | 1 (5.26) |
| **Distant Metastasis** | |
| Mx | 77 (74.76) |
| M0 | 23 (22.33) |
| M1 | 3 (2.91) |

among the heterozygote TC carriers (TC vs. TT: OR=2.31, 95%CI = 1.33–3.99, *p*-value = 0.0028). Additive model 2 showed 3.93-fold increased risk among the mutant homozygote CC carriers against reference homozygote carriers (CC vs. TT: OR=3.93, 95%CI = 1.36–11.38, *p*-value = 0.0116). According to the dominant model, 2.52-fold significant risk was identified in the breast cancer patients against the control population (TC + CC vs. TT: OR=2.52, 95%CI = 1.50–4.25, *p*-value = 0.0005). The recessive model also showed 2.77-fold increased risk in cancer patients, although the result was not statistically significant (*p*-value = 0.0545). The over-dominant model showed 1.93-fold significant risk in breast cancer patients (TC vs. TT + CC: OR=1.93, 95%CI = 1.13–3.28, *p*-value = 0.0152). The allele frequency analysis showed that mutant allele C carriers among breast cancer patients had a significantly higher odds than the wild type T allele carriers (C vs. T: OR=2.15, 95%CI = 1.41–3.26, *p*-value = 0.0003).

## Association between cervical cancer and rs3761936 polymorphism

For cervical cancer patients, additive model 1 showed 1.80-fold significantly increased risk of cancer among the heterozygote TC carrier (TC vs. TT: OR=1.80, 95%CI = 1.01–3.20, *p*-value = 0.0444). Additive model 2 showed 3.17-fold

**Table 5. Quantitative risk analysis of *DCLRE1B* rs3761936 polymorphism on breast cancer patients.**

| SNP | Model | Genotype/Allele | Cases (%) | Controls (%) | OR | 95% CI | p-value |
|---|---|---|---|---|---|---|---|
| *DCLRE1B* rs3761936 | Additive model 1 (TC vs. TT) | TT | 57 (42.22) | 70 (64.81) | 1 | | |
| | | TC | 62 (45.93) | 33 (30.56) | 2.31 | 1.33-3.99 | **0.0028** |
| | Additive model 2 (CC vs. TT) | CC | 16 (11.85) | 5 (4.63) | 3.93 | 1.36-11.38 | **0.0116** |
| | Dominant model (TC+CC vs. TT) | TT | 57 (42.22) | 70 (64.81) | 1 | | |
| | | TC+CC | 78 (57.78) | 38 (35.19) | 2.52 | 1.50-4.25 | **0.0005** |
| | Recessive model (CC vs. TT+TC) | TT+TC | 119 (88.15) | 103 (95.37) | 1 | | |
| | | CC | 16 (11.85) | 5 (4.63) | 2.77 | 0.98-7.82 | 0.0545 |
| | Over-dominant model (TC vs TT+CC) | TT+CC | 73 (54.07) | 75 (69.44) | 1 | | |
| | | TC | 62 (45.93) | 33 (30.56) | 1.93 | 1.13-3.28 | **0.0152** |
| | Allele model | T | 176 (65.18) | 173 (80.09) | 1 | | |
| | | C | 94 (34.81) | 43 (19.91) | 2.15 | 1.41-3.26 | **0.0003** |

* OR- Odd Ratio, Bold *p*-values are significant.

increased risk among the mutant homozygote CC carriers against reference homozygote carriers (CC vs. TT: OR=3.17, 95%CI = 1.05–9.55, *p*-value = 0.0403). According to the dominant model, 1.98-fold significant elevated risk was identified in the cervical cancer patients against the control population (TC+CC vs. TT: OR=1.98, 95%CI = 1.15–3.41, *p*-value = 0.0138). Both the recessive model and over-dominant models also showed increased risk in the cervical cancer patients against the reference population, although the results were not statistically significant (CC vs. TT+TC: OR=2.52, 95%CI = 0.86–7.42, *p* -value = 0.0929; TC vs. TT+CC: OR=1.57, 95%CI = 0.9–2.75, *p* -value = 0.1118). The allele frequency analysis showed that mutant allele C carriers among cervical cancer patients had a significantly higher risk than the wild type T allele carriers (C vs. T: OR=1.84, 95%CI = 1.19–2.85, *p*-value = 0.0065) ([Table 6]).

## Discussion

A key variable in the emergence of cancer is genetic polymorphism. Extensive research has been ongoing to find more precise and novel causes of cancer genetic vulnerability. Until now, more than 10,000 variants have been identified as susceptible candidates, although population-based studies sometimes show highly variable outcomes. Scientists are now attempting to identify rare variants that might have population-specific susceptibility to cancer. Among all types of cancers, breast cancer and cervical cancer are the most dominant cancers in women, especially underdeveloped countries suffer a lot from these two types of cancers due to the lack of screening and early detection or lack of awareness. In Bangladesh, female breast cancer and cervical cancer are the most common types of cancers among women, causing 12,060 deaths in 2018 [17,34–36]. According to the GLOBOCAN 2020, BC is the most common cancer that responsible for 6.2% of cancer deaths with 19% occurrence among Bangladeshi women [37]. CC is the second most common cancer with 12% occurrence as per 2018 records. Only CC caused 5,214 deaths with a total of 8,068 cases in 2018 [38]. Early detection and prevention can reduce the prevalence, incidence and mortality of the disease, but genetic susceptibility studies can play a dominant role in designing personalized medications that could prevent the occurrence of cancers due to genetic causes. Although rare variant susceptibility studies can alter the disease scenario, there are very few genetic susceptibility studies on breast cancer and cervical cancer in Bangladesh.

From the clinical point of view, patients having different genetic variations react differently to medicines. Cancer patients show high resistance to chemotherapy due to uncontrollable genetic mutations. If strong susceptible genetic loci can be identified to design personalized medicine and to adjust dose as per requirements of the patients, we will be able to reduce the therapeutic difficulties for cancer patients- both locally and globally. Moreover, if healthy persons carry a

**Table 6. Quantitative risk analysis of *DCLRE1B* rs3761936 polymorphism on cervical cancer patients.**

| SNP | Model | Genotype/Allele | Cases (%) | Controls (%) | OR | 95% CI | *p*-value |
|---|---|---|---|---|---|---|---|
| *DCLRE1B* rs3761936 | Additive model 1 (TC vs. TT) | TT | 53 (48.18) | 70 (64.81) | 1 | | |
| | | TC | 45 (40.91) | 33 (30.56) | 1.80 | 1.01-3.20 | **0.0444** |
| | Additive model 2 (CC vs. TT) | CC | 12 (10.91) | 5 (4.63) | 3.17 | 1.05-9.55 | **0.0403** |
| | Dominant model (TC+CC vs. TT) | TT | 53 (48.18) | 70 (64.81) | 1 | | |
| | | TC+CC | 57 (51.82) | 38 (35.19) | 1.98 | 1.15-3.41 | **0.0138** |
| | Recessive model (CC vs. TT+TC) | TT+TC | 98 (89.09) | 103 (95.37) | 1 | | |
| | | CC | 12 (10.91) | 5 (4.63) | 2.52 | 0.86-7.42 | 0.0929 |
| | Over-dominant model (TC vs TT+CC) | TT+CC | 65 (59.09) | 75 (69.44) | 1 | | |
| | | TC | 45 (40.91) | 33 (30.56) | 1.57 | 0.9-2.75 | 0.1118 |
| | Allele model | T | 151 (68.64) | 173 (80.09) | 1 | | |
| | | C | 69 (31.36) | 43 (19.91) | 1.84 | 1.19-2.85 | **0.0065** |

\* OR- Odd Ratio, Bold *p*-values are significant.

susceptible genetic polymorphism, they have a high chance to develop cancer in the near future. The genetic polymorphism can also remain in the family tree for generations. Early detection of potential threats can help to prevent cancer prognosis. Gene editing technologies like CRISPR are showing promising results for cancer therapy. By deleting the susceptible genetic portion of carrier human, it is possible to prevent cancer [39,40]. In Bangladesh the prevention and screening procedure for BC and CC detection has been developed over the year. Several molecular approaches have been directed toward anticancer therapy. Although gene editing technologies and personalized medication system has not established yet in the Bangladeshi anti-cancer therapeutic regime, the concepts have been investigated for many years [37,38,41]. Soon the scenario would be changed. Our present research paves the way for new treatment approaches and prevention of breast cancer and cervical cancer in Bangladesh.

*DCLRE1B* gene plays a vital role in telomere maintenance and protection during the S-phase of DNA replication. DCLRE1B protein is highly expressed in the telomere site of the chromosomes and interacts with TERF2, and involved in prophase checkpoint and prevents intra-strand DNA cross-linking by exonuclease activities. The rs3761936 of *DCLRE1B* gene is a missense variant that miscodes amino acid sequences. The altered amino acid sequence produces a different protein that might be highly susceptible to hamper DNA cross-link repair leading to the development of most cancers [19–21,26,27].

There are some genetic association studies conducted on Bangladeshi Breast cancer and cervical cancer population to evaluate susceptible single nucleotide polymorphisms of *INSIG2*, *HLA-DRB1*, *GCNT1P5, IL1β*, *IL4R*, *IL6*, *FGFR2*, *CYP1A1*, *ESR1* genes and disease outcome [33,36,42–44]. All the studies have identified significant association with BC and CC development among Bangladeshi population. Although the rs3761936 polymorphism of *DCLRE1B* gene might have a high susceptibility to developing breast cancer and cervical cancer, no specific case-control studies were conducted to evaluate the association. Only a few genetic databases showed that protein expression with the polymorphic allele of *DCLRE1B* gene was over-expressed in both breast invasive cancer cells and cervical carcinoma cells [28–30]. In a previous study, Dong et al. conducted a huge cancer association study with 344 genetic variants, of which rs3761936 polymorphism of *DCLRE1B* gene was found to be significantly associated with glioma, although the polymorphism showed protective association (OR=0.36) [34]. Another whole genome association study on rheumatoid arthritis (RA) patients of New Zealand and the United Kingdom [45] showed that the minor allele C frequency of rs3761936 SNP was significantly higher in the RA patients of both regions in comparison to the control population (*p*-value=0.001). Another

study on glioma patients also found that the minor allele frequency was significantly higher in patients than in control, and this SNP showed a lower risk for glioma [46]. GEPIA (Gene Expression Profiling Interactive Analysis) and OncoDB databases reported overexpressed *DCLRE1B* gene in both breast cancer and cervical cancer patients [47,48] as shown in Figs 3 and 4. Although the evidence was somehow confusing and contradictory, we aimed to evaluate the risk association of *DCLRE1B* rs3761936 polymorphism in Bangladeshi breast cancer and cervical cancer patients.

In this case-control study, among the breast cancer patients, 45.93% showed heterozygote TC and 11.85% showed mutant homozygote CC. In the case of allele frequency, reference allele T frequency was 65.18% and mutant allele C frequency was 34.81%. Among the cervical cancer patients, 40.91% showed heterozygote TC and 10.91% showed mutant homozygote CC. In the case of allele frequency, reference allele T frequency was 68.64% and mutant allele C frequency was 31.36%. Minor allele frequency was significantly higher in both the breast cancer and cervical cancer patients than in the control population. In genetic model analysis of breast cancer patients, additive model 1 showed 2.31-fold (OR=2.31) significantly increased risk among the heterozygote TC carriers (*p*-value=0.0028). 3.93-fold increased risk was identified in additive model 2 among the mutant homozygote CC carriers (OR=3.93; *p*-value=0.0116). The dominant model showed 2.52-fold (OR=2.52) significant risk in the breast cancer patients against the control population (*p*-value=0.0005). The recessive model also showed increased risk, but the result was not statistically significant (*p*-value=0.0545). The over-dominant model also showed 1.93-fold significant risk in breast cancer patients (OR=1.93; *p*-value=0.0152). The mutant allele C carriers in the breast cancer patients had a significantly higher risk than the wild-type T allele carriers (*p*-value=0.0003).

For cervical cancer patients, similar results were observed. Additive model 1 showed 1.80-fold significantly increased risk in the heterozygote TC carrier (OR=1.80; *p*-value=0.0444). 3.17-times odds were identified in additive model 2 among the mutant homozygote CC carriers (OR=3.17; *p*-value=0.0403). The dominant model showed 1.98-fold significant risk in the cervical cancer patients against the control population (OR=1.98; *p*-value=0.0138). Both the recessive

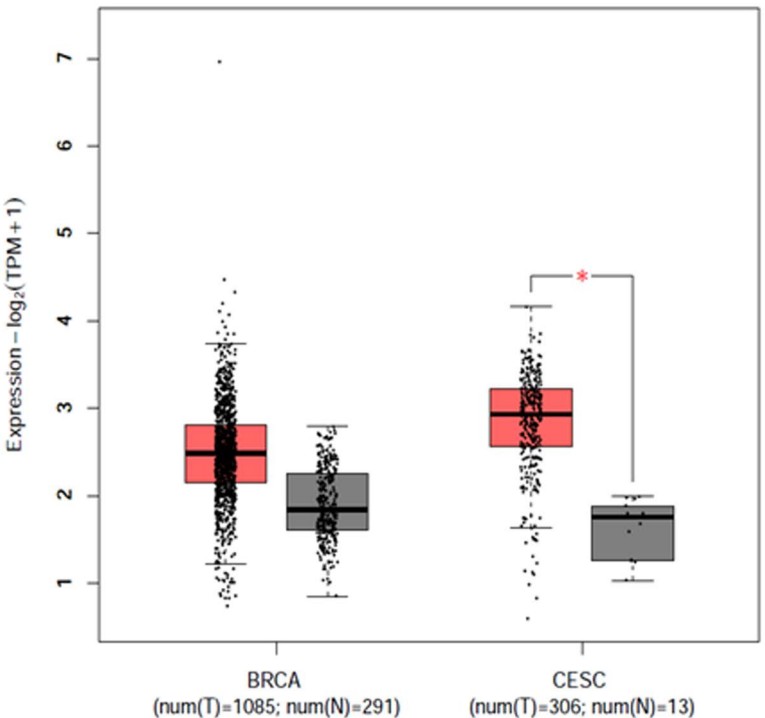

**Fig 3. GEPIA2 database on the genetic expression of *DCLRE1B* gene in breast cancer and cervical cancer patients.**

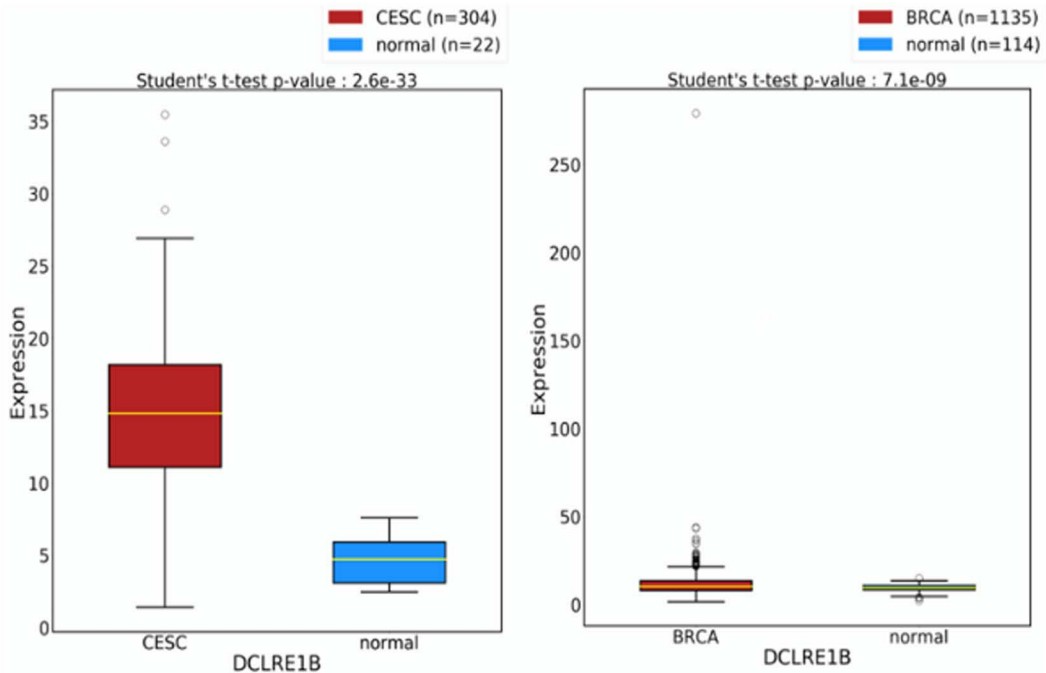

**Fig 4. OncoDB report on the genetic expression of *DCLRE1B* gene in breast cancer and cervical cancer patients.**

model and over-dominant models showed increased risk in the cervical cancer patients against the reference population, although the results were not statistically significant ($p$-value>0.05). The allele frequency analysis showed that mutant allele C carriers had a significantly higher risk than the wild-type T allele carriers ($p$-value = 0.0065).

The presence of mutant allele among both cancer patients were significantly higher than the control population that indicates the vital association of this SNP in the cancer development. A person carrying the mutant allele will be more susceptible to develop cancer than the wild type allele carrier as our results showed significantly high-risk association among the cancer patients carrying this allele.

Although this is the first case-control study of *DCLRE1B* rs3761936 polymorphism with the risk of breast and cervical cancer, there were some limitations that should be mentioned. The sample size of the patients was limited. For genetic association studies, the larger the sample size, the stronger the consistency and stability of the results. Some of our results were not statistically significant may be due to the smaller sample size. Apart from the smaller sample size, we were able to report consistency with the HWE constant. Except for the mentioned limitations, our present case-control study confirmed that *DCLRE1B* rs3761936 polymorphism has a strong association with breast cancer and cervical cancer risk in Bangladeshi women.

## Conclusion

*DCLRE1B* rs3761936 polymorphism is strongly associated with a significantly increased risk of breast cancer and cervical cancer in Bangladeshi women. This result highlights the possibility that SNP rs3761936 might be associated with the other cancer forms and would be a potential therapeutic target for all types of cancers. The variation of the chemotherapeutic drug resistance among the polymorphism carriers should be thoroughly investigated to design personalized medication strategy. Our current findings revealed a promising source in the field of cancer therapeutics that can become a sustainable solution to cancer. However, a single study cannot establish firm evidence as genetic variations among different

ethnicities are highly noticeable. So, further studies with larger samples on the other ethnic population should be conducted to confirm and re-evaluate our findings.

## Supporting information

**S1 Table. Primer sequences used for amplification.**
(DOCX)

**S2 Table. Distribution of genotype and allelic frequencies of *DCLRE1B* rs3761936 polymorphism between cancer patients and control population.**
(DOCX)

**S1 File. Supporting data set.**
(ZIP)

## Acknowledgments

The authors would like to thank the Department of Pharmacy, Noakhali Science and Technology University, for their assistance and support during the project. Cordial thanks to the National Institute of Cancer Research and Hospital (NICRH), Dhaka, for their valuable contribution.

## Author contributions

**Conceptualization:** Mohammad Safiqul Islam.

**Data curation:** Sarah Jafrin, Md. Abdul Aziz, Md Abdul Barek.

**Formal analysis:** Sarah Jafrin, Mohammad Safiqul Islam.

**Funding acquisition:** Mohammad Safiqul Islam.

**Investigation:** Sarah Jafrin, Nura Ershad Naznin.

**Methodology:** Sarah Jafrin, Md. Abdul Aziz.

**Project administration:** Md Abdul Barek, Mohammad Safiqul Islam.

**Software:** Md. Abdul Aziz, Md Abdul Barek, Md. Sharif Reza, Mohammad Safiqul Islam.

**Supervision:** Mohammad Safiqul Islam.

**Validation:** Sarah Jafrin, Md. Abdul Aziz, Md. Sharif Reza, Nura Ershad Naznin.

**Visualization:** Md. Abdul Aziz, Md Abdul Barek, Md. Sharif Reza, Nura Ershad Naznin.

**Writing – original draft:** Sarah Jafrin, Md Abdul Barek, Md. Sharif Reza, Nura Ershad Naznin.

**Writing – review & editing:** Md. Abdul Aziz, Md Abdul Barek, Mohammad Safiqul Islam.

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
