## [Decision Letter · Decision Letter 0]

11 Mar 2024

Dear Dr. Islam,

Thank you for submitting your manuscript to PLOS ONE. After careful consideration, we feel that it has merit but does not fully meet PLOS ONE’s publication criteria as it currently stands. Therefore, we invite you to submit a revised version of the manuscript that addresses the points raised during the review process.

We look forward to receiving your revised manuscript.

Kind regards,

Milad Khorasani, PhD

Academic Editor

PLOS ONE

Journal Requirements:

Did you know that depositing data in a repository is associated with up to a 25% citation advantage (https://doi.org/10.1371/journal.pone.0230416)? If you’ve not already done so, consider depositing your raw data in a repository to ensure your work is read, appreciated and cited by the largest possible audience. You’ll also earn an Accessible Data icon on your published paper if you deposit your data in any participating repository (https://plos.org/open-science/open-data/#accessible-data).

"The Research Cell, Noakhali Science and Technology University, Noakhali-3814, Bangladesh, funded this study partially (NSTU/RC/20/C-86), and no other public or private logistic funding was provided."

4. Please note that funding information should not appear in the Acknowledgments section or other areas of your manuscript. We will only publish funding information present in the Funding Statement section of the online submission form. Please remove any funding-related text from the manuscript. 

5. In this instance it seems there may be acceptable restrictions in place that prevent the public sharing of your minimal data. However, in line with our goal of ensuring long-term data availability to all interested researchers, PLOS’ Data Policy states that authors cannot be the sole named individuals responsible for ensuring data access (http://journals.plos.org/plosone/s/data-availability#loc-acceptable-data-sharing-methods).

Reviewers' comments:

Reviewer's Responses to Questions

**Comments to the Author**

1. Is the manuscript technically sound, and do the data support the conclusions?

Reviewer #1: Yes

Reviewer #2: Yes

2. Has the statistical analysis been performed appropriately and rigorously?

Reviewer #1: Yes

Reviewer #2: Yes

3. Have the authors made all data underlying the findings in their manuscript fully available?

Reviewer #1: Yes

Reviewer #2: Yes

4. Is the manuscript presented in an intelligible fashion and written in standard English?

Reviewer #1: Yes

Reviewer #2: Yes

Reviewer #1: This is an interesting paper on the association of missense variant DCLRE1B rs3761936 with breast and cervical cancer. However, due to the design of the study, it is difficult to establish causation, as the authors commented in the discussion

The study is based on a researcher-designed interview-administered questionnaire, used to collect the required information. I have not seen any reports on validity and reliability

Please highlight the novelty of this study in the conclusions. Discuss the clinical relevance and applicability of the study results locally & globally.

Two-sided unpaired t-tests or Independent Samples t Test???

The logistic regression analysis is not well described, specifically with respect to how the multivariable analysis was performed and which variables were selected for inclusion as covariates in the models

Describe the variables used in the study in the method section

State the inclusion and exclusion criteria of the case and control groups

Use more odds than risk in OR interpretation

What is missing in the discussion is that the authors should not only compare their studies with similar studies but also discuss the meaning of their findings within the context of the Bangladesh literature, which I think there are a lot. Secondly, the authors need to discuss the policy, practice and research implications of their findings. e.g what does their finding mean regarding the prevention of breast and cervical cancer risk guidelines in Bangladesh?

Reviewer #2: These are the following observation:

1. Is it necessary to write The in the starting of the title? I think, word THE is not necessary.

2. The primer of the gene is missing. Please add the primer.

3. Please mention the name of the thermocycler with country of origin.

4. In Table, the authors have mentioned the mean values but without SD. It is mandatory to mention. Please mention SD with each mean values.

5. Statistical analysis is incomplete. Please write according to the analysis.

**Do you want your identity to be public for this peer review?** For information about this choice, including consent withdrawal, please see our Privacy Policy

Reviewer #1: No

Reviewer #2: **Yes: ** Dr. Md. Abdullah Yusuf

---

## [Author Response · Author response to Decision Letter 1]

17 Apr 2024

We are thankful to the editor and the reviewers for their crucial comments on our manuscript. We hope these comments will help us improve it.

Journal Academic Editor

Response: All the requirements are fulfilled as per journal requirements.

2. Consider depositing your raw data in a repository to ensure your work is read, appreciated and cited by the largest possible audience. You’ll also earn an Accessible Data icon on your published paper if you deposit your data in any participating repository.

Response: This suggestion is crucial. We have not deposited these data in any repository, but we added our data as an EXCEL file as supporting materials.

Response: The statement is included in the Funding information file.

4. Please note that funding information should not appear in the Acknowledgments section or other areas of your manuscript. We will only publish funding information present in the Funding Statement section of the online submission form. Please remove any funding-related text from the manuscript.

Response: Removed from the manuscript and saved in a ‘financial information’ named file.

5. In this instance it seems there may be acceptable restrictions in place that prevent the public sharing of your minimal data. However, in line with our goal of ensuring long-term data availability to all interested researchers, PLOS’ Data Policy states that authors cannot be the sole named individuals responsible for ensuring data access (http://journals.plos.org/plosone/s/data-availability#loc-acceptable-data-sharing-methods).

Response: Author information are shared. All the related information will be shared.

6. Please include captions for your Supporting Information files at the end of your manuscript, and update any in-text citations to match accordingly.

Response: Done.

Reviewer 1

1. Please highlight the novelty of this study in the conclusions. Discuss the clinical relevance and applicability of the study results locally & globally.

Response: See the conclusion and discussion section with comments.

2. Two-sided unpaired t-tests or Independent Samples t Test???

The logistic regression analysis is not well described, specifically with respect to how the multivariable analysis was performed and which variables were selected for inclusion as covariates in the models

Response: T-tests and other variables are explained in the statistical analysis section. Check comment.

3. Describe the variables used in the study in the method section

State the inclusion and exclusion criteria of the case and control groups

Response: Done. Inclusion and exclusion criteria are added in the method section.

4. Use more odds than risk in OR interpretation

Response: Resolved the issue

5. What is missing in the discussion is that the authors should not only compare their studies with similar studies but also discuss the meaning of their findings within the context of the Bangladesh literature, which I think there are a lot. Secondly, the authors need to discuss the policy, practice and research implications of their findings. e.g what does their finding mean regarding the prevention of breast and cervical cancer risk guidelines in Bangladesh?

Response: Added more information in relation to Bangladeshi BC and CC cancer guidelines. Tried to explain all the points as suggested. Please see the comment section of the discussion.

Reviewer 2

1. Is it necessary to write The in the starting of the title? I think, word THE is not necessary.

Response: ‘The’ is removed from the title.

2. The primer of the gene is missing. Please add the primer.

Response: Please check the supporting table 1.

3. Please mention the name of the thermocycler with country of origin.

Response: Mentioned in the method section.

4. In Table, the authors have mentioned the mean values but without SD. It is mandatory to mention. Please mention SD with each mean values

Response: SD values are added to the mean values. Check Table 2.

5. Statistical analysis is incomplete. Please write according to the analysis.

Response: Statistical analysis is explained as suggested according to the analysis.

---

## [Decision Letter · Decision Letter 1]

23 Jun 2024

Dear Dr. Islam,

Thank you for submitting your manuscript to PLOS ONE. After careful consideration, we feel that it has merit but does not fully meet PLOS ONE’s publication criteria as it currently stands. Therefore, we invite you to submit a revised version of the manuscript that addresses the points raised during the review process.

We look forward to receiving your revised manuscript.

Kind regards,

Milad Khorasani, PhD

Academic Editor

PLOS ONE

Reviewers' comments:

Reviewer's Responses to Questions

**Comments to the Author**

Reviewer #1: (No Response)

Reviewer #2: All comments have been addressed

2. Is the manuscript technically sound, and do the data support the conclusions?

Reviewer #1: Yes

Reviewer #2: Partly

3. Has the statistical analysis been performed appropriately and rigorously?

Reviewer #1: No

Reviewer #2: Yes

4. Have the authors made all data underlying the findings in their manuscript fully available?

Reviewer #1: (No Response)

Reviewer #2: Yes

5. Is the manuscript presented in an intelligible fashion and written in standard English?

Reviewer #1: (No Response)

Reviewer #2: Yes

Reviewer #1: Dear Editor

My comments 1, 4, and 5 have not been answered

Also, track changes or highlight the responses to the reviewer's comments

Reviewer #2: The authors have tested less number of control than case group. How they match this?

The propensity scoring analysis is missing to compare the group. It is very difficult to say that there is a genetic association unless otherwise properly matched the 2 groups. Please explain why this happen.

The methodology section is poorly mentioned about the procedure of the genetic tests. Explain in details.

The discussion section needs more elaboration before concretely said that there is a genetic association.

**Do you want your identity to be public for this peer review?** For information about this choice, including consent withdrawal, please see our Privacy Policy

Reviewer #1: No

Reviewer #2: **Yes: ** Dr. Md. Abdullah Yusuf

---

## [Author Response · Author response to Decision Letter 2]

25 Jul 2025

PONE-D-23-33713R1

Association of missense variant DCLRE1B rs3761936 with breast and cervical cancer risk - A case-control study

We are thankful to the editor and the reviewers for their crucial comments on our manuscript. We hope these comments will help us improve it.

Reviewer 1

My comments 1, 4, and 5 have not been answered

Also, track changes or highlight the responses to the reviewer's comments.

Response:

All the comments by the reviewer were considered and revised accordingly. Please see the track changes and comments of the edited manuscript with track changes.

Reviewer 2

The authors have tested less number of control than case group. How they match this?

The propensity scoring analysis is missing to compare the group. It is very difficult to say that there is a genetic association unless otherwise properly matched the 2 groups. Please explain why this happen.

The methodology section is poorly mentioned about the procedure of the genetic tests. Explain in detail.

The discussion section needs more elaboration before concretely said that there is a genetic association.

Response:

Reviewer 2 The authors have tested less number of control than case group. How they match this?

The propensity scoring analysis is missing to compare the group. It is very difficult to say that there is a genetic association unless otherwise properly matched the 2 groups. Please explain why this happen.

The methodology section is poorly mentioned about the procedure of the genetic tests. Explain in detail.

The discussion section needs more elaboration before concretely said that there is a genetic association.

Response:

We used a single control group (108) for two case groups (135 BC and 110 CC). Due to COVID-19 pandemic situation, our control sample collection was limited. Although we tested less number of controls against the case groups during genetic association analysis, we maintained the ratios within the acceptable range and heterogeneity, along with other statistical variables were significantly controlled. Valid estimates of the odds were obtained by following appropriate sample selection processes.

The variant under investigation is inherited from parents, so an individual’s genotype is set at conception and is generally independent of lifestyle or socio‑demographic characteristics. Because such germline alleles are passed down randomly, they are not expected to correlate with non‑genetic confounders. For this reason, propensity‑score matching is rarely applied in genetic case–control research and is not yet a routine tool for controlling confounding in this field. Instead, we accounted for potential differences between cases and controls in age and other risk factors using multivariable logistic regression models. This widely accepted approach preserves statistical power and avoids the information loss that can occur when matching on many variables. We add a statement regarding this before the conclusion section in the manuscript.

The methodology section is properly explained as per the reviewer’s suggestion. Please see the Methodology section of the revised manuscript. (Highlighted portion)

The Discussion section is elaborated as per the reviewer’s comment. (Highlighted portion)

---

## [Decision Letter · Decision Letter 2]

18 Aug 2025

Association of missense variant DCLRE1B rs3761936 with breast and cervical cancer risk - A case-control study

PONE-D-23-33713R2

Dear Dr. Safiqul Islam,

We’re pleased to inform you that your manuscript has been judged scientifically suitable for publication and will be formally accepted for publication once it meets all outstanding technical requirements.

Kind regards,

Milad Khorasani, PhD

Academic Editor

PLOS ONE

Additional Editor Comments (optional):

Reviewers' comments:

Reviewer's Responses to Questions

**Comments to the Author**

Reviewer #2: All comments have been addressed

2. Is the manuscript technically sound, and do the data support the conclusions?

Reviewer #2: Yes

3. Has the statistical analysis been performed appropriately and rigorously?

Reviewer #2: Yes

4. Have the authors made all data underlying the findings in their manuscript fully available?

Reviewer #2: Yes

5. Is the manuscript presented in an intelligible fashion and written in standard English?

Reviewer #2: Yes

Reviewer #2: The correction are properly done, however, the comparison of case and control group values should be compared and the p values are missing in the tables.

**Do you want your identity to be public for this peer review?** For information about this choice, including consent withdrawal, please see our Privacy Policy

Reviewer #2: **Yes: ** Md Abdullah Yusuf

---

## [Editor Report · Acceptance letter]

PONE-D-23-33713R2

PLOS ONE

Dear Dr. Islam,

I'm pleased to inform you that your manuscript has been deemed suitable for publication in PLOS ONE. Congratulations! Your manuscript is now being handed over to our production team.

Kind regards,

on behalf of

Dr. Milad Khorasani

Academic Editor

PLOS ONE